# Learning versus Refutation in Noninteractive Local Differential Privacy

**Alexander Edmonds**
University of Toronto
alex.edmonds@utoronto.ca

**Aleksandar Nikolov**
University of Toronto
anikolov@cs.toronto.edu

**Toniann Pitassi**
Columbia University & University of Toronto
toni@cs.toronto.edu

## Abstract

We study two basic statistical tasks in non-interactive local differential privacy (LDP): *learning* and *refutation*: learning requires finding a concept that best fits an unknown target function (from labelled samples drawn from a distribution), whereas refutation requires distinguishing between data distributions that are well-correlated with some concept in the class, versus distributions where the labels are random. Our main result is a complete characterization of the sample complexity of agnostic PAC learning for non-interactive LDP protocols. We show that the optimal sample complexity for any concept class is captured by the approximate $\gamma_2$ norm of a natural matrix associated with the class. Combined with previous work, this gives an *equivalence* between agnostic learning and refutation in the agnostic setting.

## 1 Introduction

We study two related basic statistical tasks, *learning* and *refutation*, in the setting of distributed data, and under strong privacy constraints. For both tasks, we have an unknown distribution $\lambda$ on labeled data points in the universe $\mathcal{U} \times \{\pm 1\}$, and we receive samples from $\lambda$. We are also given a concept class $\mathcal{C} \subseteq \{\pm 1\}^{\mathcal{U}}$, which, hopefully, is capable of capturing the labels given by $\lambda$. We define our two tasks as follows.

- *Learning* requires finding a concept that best fits $\lambda$. I.e., using the usual binary loss function $L_\lambda(h) = \underset{(a,b)\sim\lambda}{\mathbb{E}} [\mathbb{I}[h(a) \neq y]]$, the goal of agnostic learning with accuracy $\alpha$ is to produce some $h$ which, with probability $1 - \beta$, satisfies $L_\lambda(h) \leq \min_{c\in\mathcal{C}} L_\lambda(c) + \alpha$.
  If an algorithm solves this problem for any distribution $\lambda$, then we say it $(\alpha, \beta)$-learns $\mathcal{C}$ agnostically.

- *Refutation* requires distinguishing between data distributions $\lambda$ that are well correlated with some concept $c \in \mathcal{C}$, vs. data distributions where the labels are random. I.e., the goal of agnostic refutation with accuracy $\alpha$ is to distinguish, with probability $1 - \beta$, between the following two cases: (i) $\min_{c\in\mathcal{C}} L_\lambda(c) \leq \frac{1}{2} - \alpha$ versus (ii) for all $h \in \{\pm 1\}^{\mathcal{U}}$, $L_\lambda(h) = \frac{1}{2}$. If an algorithm solves this problem for any distribution $\lambda$, then we say it $(\alpha, \beta)$-refutes $\mathcal{C}$ agnostically.

The definition of agnostic learning above is classical. Refutation is a more recent notion, and was studied by Kothari and Livni [2018] (and in a realizable setting by Vadhan [2017]), where it was

shown that computationally efficient refutation is equivalent to computationally efficient agnostic learning. Refutation captures an extreme version of the problem of evaluating the choice of model in supervised learning, i.e., of estimating the best achievable loss $\min_{c \in \mathcal{C}} L_\lambda(c)$ by the concept class $\mathcal{C}$. While agnostic learning is well-defined for any concept class, it is less meaningful when the best achievable loss is trivially large, which may be an indication that we need to choose a different model, i.e., a different concept class. For this reason, ideally we would like our learning algorithm to also tell us what loss it is able to achieve. Refutation is a more basic version of this problem, in which we merely want to distinguish data distributions for which our model is good from distributions with random labels, for which no model can achieve good results. Certainly being able to solve the refutation problem is at least as hard as estimating $\min_{c \in \mathcal{C}} L_\lambda(c)$.

In this paper, we study learning and refutation in the model of *non-interactive local differential privacy* (LDP) [Kasiviswanathan et al., 2008]. LDP applies in a distributed setting in which each data point represents one person, and, in order to protect privacy, the person retains ownership of their data point. In particular, the data is never centrally collected, and, instead, the data owners communicate differentially private randomized message to a central server. The differential privacy [Dwork et al., 2006] constraint ensures that the distribution on messages sent by one participant does not change dramatically if that participant's data point is changed. Thus, the central server or an outside observer cannot learn much about any particular data point, guaranteeing a strong form of privacy protection (as long as the privacy parameter is small enough). Nevertheless, with enough participants, the combination of all private messages can reveal enough statistical information in aggregate in order to solve a statistical task, such as learning. LDP is the model of choice of many industrial deployments of differential privacy [Erlingsson et al., 2014, Thakurta et al., 2017, Apple, 2016, Ding et al., 2017]. Here we focus on *non-interactive* LDP protocols, i.e., protocols in which each participant simultaneously sends a single message to the server. Non-interactive protocols are much easier to implement than multi-round interactive protocols, particularly considering the large number of data points which are typically necessary for LDP to be useful.

Our main goal is to characterize, for any given concept class $\mathcal{C}$, the sample complexity of learning and refutation under the constraints of non-interactive LDP. Moreover, we aim to understand how these two problems are related to each other.

In many settings, it is trivial to take an algorithm for learning and use it to obtain an algorithm for refutation, by executing the learning algorithm for accuracy $\alpha/4$, and estimating the loss of the returned hypothesis within $\alpha/4$. A converse of this simple reduction was established by Kothari and Livni [2018], and by Vadhan [2017]. Unfortunately, neither of these reductions applies to the setting of non-interactive LDP, since they rely on interacting with the distribution $\lambda$ adaptively.

This leaves open the question of whether or not learning and refutation in the non-interactive LDP setting are equivalent tasks with respect to sample complexity. In the forward (typically easier) direction, can non-interactive learning algorithms solve non-interactive refutation (by estimating the minimum loss achievable by the model) without a significant increase in sample complexity, and conversely, can the reductions from refutation to learning from Kothari and Livni [2018] and Vadhan [2017] be extended to the setting of non-interactive protocols?

We note that, by the equivalence proved in Kasiviswanathan et al. [2011] between LDP and the statistical queries (SQ) model of Kearns [1993], this also means that the relationship between the query complexity of non-adaptive SQ learning versus refutation is open. Similarly, all our results extend to the non-adaptive SQ model. Adaptive SQ learning has been characterized by Feldman [2017], and this in turn implies the same characterization for sequential LDP (LDP protocols in which each participant sends one message, which can depend on the messages of previous participants).

An overview of our main results follows. The derivation of our results will be presented in Section 2 for the agnostic setting and in Section 3 for realizable. See Appendix A for notation and standard definitions. See Appendix E for a discussion of open problems.

## 1.1 Characterization of agnostic learning

Our first theorem resolves the open problem for agnostic learning, by showing that non-interactive LDP learning and refutation are equivalent (up to a logarithmic approximation) in the agnostic setting. We do so by the following theorem, which gives a characterization of the sample complexity of both

problems in terms of the approximate $\gamma_2$ norm of a natural matrix associated with the concept class $\mathcal{C} \subseteq \{\pm 1\}^{\mathcal{U}}$.

**Theorem 1.** *Let $\mathcal{C} \subseteq \{\pm 1\}^{\mathcal{U}}$ be a finite concept class with concept matrix $W \in \{\pm 1\}^{\mathcal{C} \times \mathcal{U}}$, as given by Definition 3. Let $\varepsilon > 0$, $\alpha, \beta \in (0, 1/2]$. Then, to either $(2\alpha, \beta)$-learn $\mathcal{C}$ agnostically, or $(2\alpha, \beta)$-refute $\mathcal{C}$ agnostically under non-interactive $\varepsilon$-LDP, it suffices to have a sample of size*

$$n = O\left( \frac{\gamma_2(W, \alpha)^2 \cdot \log(|\mathcal{C}|/\beta)}{\varepsilon^2 \alpha^2} \right).$$

*Conversely, for $\beta = \frac{1}{2} - \Omega(1)$, for some $\alpha' = \Omega\left( \frac{\alpha}{\log(1/\alpha)} \right)$, the number of samples required to either $(\alpha', \beta)$-learn agnostically or $(\alpha', \beta)$-refute $\mathcal{C}$ agnostically under non-interactive $\varepsilon$-LDP is at least*

$$n = \Omega\left( \frac{(\gamma_2(W, \alpha) - 1)^2}{\varepsilon^2 \alpha^2} \right).$$

In Theorem 1, we denote by $\gamma_2(W, \alpha)$ the approximate $\gamma_2$ norm of the matrix $W$, i.e., the minimum $\gamma_2$ norm of a matrix that approximates $W$ up to an additive $\alpha$ entrywise. (For a definition of the $\gamma_2$ norm, see Appendix A.) The theorem shows that the sample complexity of both learning and refutation under non-interactive LDP can be characterized in terms of $\gamma_2(W, \alpha)$. Moreover, the sample complexities of both problems are equal, up to a factor $O(\log(1/\alpha))$ loss in the accuracy parameter, and a factor $O(\log |\mathcal{C}|)$ loss in the sample complexity.

The main new result in Theorem 1 is the lower bound on the sample complexity of learning. The upper bound for both learning and refutation, as well as the lower bound for refutation were previously shown by Edmonds et al. [2019]. We close the gap by proving the lower bound for learning. Combined with the previous results, this proves the equivalence of non-interactive LDP sample complexity for learning and refutation. As in the previous proofs, we prove the lower bound for agnostic learning via the (dual formulation) of the approximate $\gamma_2$ norm. In order to give a family of distributions that is hard against learning algorithms, we define a new *difference* matrix, $D$, associated with the concept class $\mathcal{C}$, which is more suitable for the learning lower bound. Then we show that $\gamma_2(D, \alpha)$ and $\gamma_2(W, \alpha)$ are approximately equal.

## 1.2 Characterization of realizable refutation

The results above do not apply to the realizable setting, in which the underlying distribution $\lambda$ on $\mathcal{U} \times \{\pm 1\}$ is guaranteed to be labelled by a concept $c \in \mathcal{C}$, i.e., $L_\lambda(c) = 0$. In particular, the lower bounds we prove in terms of the approximate $\gamma_2$ norm utilize distributions that may not be realizable. It is worth noting that some concept classes may require exponentially fewer samples to learn in the realizable case than to learn agnostically, as may be seen with Kearns [1993] and Feldman [2009].

While we are not able to characterize realizable learning, we give a characterization of a realizable analog of the refutation problem, and show that realizable learning is no harder than realizable refutation. In our formulation of realizable refutation with accuracy $\alpha$, we are given samples from some distribution $\lambda$ over $\mathcal{U} \times \{\pm 1\}$, and the goal is to distinguish the cases:

- $\min_{c \in \mathcal{C}} L_\lambda(c) = 0$, i.e, some concept in $\mathcal{C}$ exactly gives the labels under $\lambda$;
- for all $h \in \{\pm 1\}^{\mathcal{U}}$, $L_\lambda(h) \geq \alpha$.

The $\alpha = \frac{1}{2}$ case is equivalent to the definition of refutation introduced by Vadhan [2017].

Daniely and Feldman [2018] showed that (for $\mathcal{C}$ closed under negation) the sample complexity of realizable learning under non-interactive LDP is bounded from below by the margin complexity of $\mathcal{C}$. They left open the question whether one can prove a matching upper bound. This question was resolved in the negative by Dagan and Feldman [2019]. The problem of characterizing the sample complexity of realizable learning under non-interactive LDP thus remains open.

In this work, we give a non-interactive LDP protocol which may be applied towards both realizable learning and realizable refutation. This gives a sample complexity upper bound for these problems in terms of a new efficiently computable quantity $\eta(\mathcal{C}, \alpha)$ that we define. This quantity combines elements of both the $\gamma_2$ norm and of margin complexity, and is sandwiched between them. Further, we derive a lower bound for realizable refutation in terms of $\eta(\mathcal{C}, \alpha)$, showing that our protocol is

nearly optimal for realizable refutation, and that the sample complexity of realizable refutation is an upper bound on the sample complexity of realizable learning under non-interactive LDP. Our main theorem for realizable learning is stated next. See Section 3 for the definition of $\eta(\mathcal{C}, \alpha)$.

**Theorem 2.** *Let $\mathcal{C} \subseteq \{\pm 1\}^{\mathcal{U}}$ be a finite concept class. Let $\varepsilon > 0$, $\alpha, \beta \in (0, 1/2]$. Then, to either $(2\alpha, \beta)$-learn $\mathcal{C}$ realizably, or $(2\alpha, \beta)$-refute $\mathcal{C}$ realizably under non-interactive $\varepsilon$-LDP, it suffices to have a sample of size*

$$n = O\left(\frac{\eta(\mathcal{C}, \alpha)^2 \cdot \log(|\mathcal{C}|/\beta)}{\varepsilon^2 \alpha^2}\right).$$

*Conversely, for $\beta = \frac{1}{2} - \Omega(1)$, for some $\alpha' = \Omega\left(\frac{\alpha}{\log(1/\alpha)}\right)$, the number of samples required to $(\alpha', \beta)$-refute $\mathcal{C}$ realizably under non-interactive $\varepsilon$-LDP is at least*

$$n = \Omega\left(\frac{\eta(\mathcal{C}, \alpha)^2}{\varepsilon^2 \alpha^2}\right).$$

## 2 Refutation versus Learning: Agnostic Case

As mentioned in the introduction, Edmonds et al. [2019] (Theorems 24 and 25) gave sample complexity upper bounds for both agnostic learning and refutation for non-interactive LDP in terms of the approximate $\gamma_2$, as well as a nearly tight lower bound for agnostic refutation. However, it left open the question of lower bounds for agnostic learning under non-interactive LDP. Our main theorem, stated next, resolves this by giving a nearly tight lower bound in terms of the approximate $\gamma_2$ norm of the natural matrix associated with $\mathcal{C}$. Theorem 1 thus follows by combining Theorems 24 and 25 from Edmonds et al. [2019] together with Theorem 4.[1]

**Definition 3.** *Let $\mathcal{C} \subseteq \{\pm 1\}^{\mathcal{U}}$ be a concept class. The concept matrix $W_{\mathcal{C}} \in \{\pm 1\}^{\mathcal{C} \times \mathcal{U}}$ of $\mathcal{C}$ is the matrix with entries given by $w_{c,a} = c(a)$.*

**Theorem 4.** *Let $\mathcal{C} \subseteq \{\pm 1\}^{\mathcal{U}}$ be a concept class with concept matrix $W \in \{\pm 1\}^{\mathcal{C} \times \mathcal{U}}$ as given by Definition 3. Let $\varepsilon > 0$, $\alpha, \beta \in (0, 1]$. Assume $\beta = \frac{1}{2} - \Omega(1)$. Then, for some $\alpha' = \Omega\left(\frac{\alpha}{\log(1/\alpha)}\right)$, under non-interactive $\varepsilon$-LDP, the number of samples required to $(\alpha', \beta)$-learn $\mathcal{C}$ agnostically is at least*

$$n = \Omega\left(\frac{(\gamma_2(W, \alpha) - 1)^2}{\varepsilon^2 \alpha^2}\right).$$

### 2.1 Difference matrix

Theorem 4 is given in terms of the concept matrix associated with the concept class; however, our proof of this result will focus instead on the *difference* matrix associated with the concept class, defined below.

**Definition 5.** *The difference matrix of a concept class $\mathcal{C} : \mathcal{U} \to \{\pm 1\}$ is the matrix $D \in \{\pm 1\}^{\mathcal{C}^2 \times \mathcal{U}}$ with entries given, for $c, c' \in \mathcal{C}, a \in \mathcal{U}$, by*

$$d_{(c,c'),a} = \frac{1}{2}\left(c(a) - c'(a)\right) = \begin{cases} 0 & \text{if } c(a) = c'(a) \\ -1 & \text{if } c(a) = -1, c'(a) = +1 \\ +1 & \text{if } c(a) = +1, c'(a) = -1. \end{cases} \tag{1}$$

The difference matrix is one of the key ideas that enables the proof of Theorem 4. We will use a dual formulation of $\gamma_2(D, \alpha)$ to construct pairs of hard distributions for our lower bound, each pair corresponding to a pair of concepts $c, c' \in \mathcal{C}$. The structure of the difference matrix will help us ensure that no correct agnostic learning algorithm can output, with high probability, the same hypothesis for both distributions in a pair. It is not apparent how to guarantee this property when working directly with the concept matrix $W$. Nevertheless, the following lemma shows that $\gamma_2(D, \alpha)$ and $\gamma_2(W, \alpha)$ are essentially the same. (See Appendix B for the proof.)

---

[1]While Theorems 24 and 25 from Edmonds et al. [2019] are stated in terms of agnostic learning, their definition of agnostic learning is non-standard and requires the learner to output a hypothesis as well as the loss it achieves. Thus the upper bounds hold for the standard definition of agnostic learning, while the lower bound only holds for refutation.

**Lemma 6.** *Let $\mathcal{C}$ be a concept class with concept matrix $W \in \mathbb{R}^{\mathcal{C} \times \mathcal{U}}$ and difference matrix $D \in \mathbb{R}^{\mathcal{C}^2 \times \mathcal{U}}$. Then $\gamma_2(D, \alpha) \leq \gamma_2(W, \alpha)$. Conversely, $\gamma_2(W, \alpha) \leq 2\gamma_2(D, \alpha/2) + 1$, and if $\mathcal{C}$ is closed under negation then $\gamma_2(W, \alpha) \leq \gamma_2(D, \alpha)$.*

The next Lemma is the same as Theorem 4, but with $W$ replaced by the difference matrix $D$. Theorem 4 is an immediate consequence of Lemma 7, together with Lemma 6.

**Lemma 7.** *Let $\mathcal{C} \subseteq \{\pm 1\}^{\mathcal{U}}$ be a concept class with concept matrix $D \in \{\pm 1\}^{\mathcal{C} \times \mathcal{U}}$ as given by Definition 3. Then Theorem 4 holds with $W$ replaced by the difference matrix, $D$.*

The rest of this section is devoted to the proof of Lemma 7.

## 2.2 KL-divergence bound

For a mechanism $\mathcal{M} : \mathcal{X}^n \to \mathcal{Z}$ in the local model, and a probability distribution $\nu$ on $\mathcal{X}^n$, we use $\mathcal{T}_{\mathcal{M}}(\nu)$ to denote the distribution of the mechanism's transcript when its input is sampled from $\nu$.

**Lemma 8.** *[Edmonds et al., 2019] Let $\varepsilon \in (0, 1]$, and let $\mathcal{M} : \mathcal{X} \to \mathcal{Z}$ be a non-interactive $\varepsilon$-LDP protocol. Then, for families $\{\lambda_1, \ldots, \lambda_k\}$ and $\{\mu_1, \ldots, \mu_k\}$ of distributions on $\mathcal{X}$, together with a distribution $\pi$ over $[k]$,*

$$\mathbb{E}_{V \sim \pi}\left[D_{KL}(\mathcal{T}_{\mathcal{M}}(\lambda_V^n) \| \mathcal{T}_{\mathcal{M}}(\mu_V^n))\right] \leq O(n\varepsilon^2) \cdot \max_{f \in \mathbb{R}^{\mathcal{X}} : \|f\|_\infty \leq 1} \mathbb{E}_{V \sim \pi}\left[\left(\mathbb{E}_{x \sim \lambda_V}[f_x] - \mathbb{E}_{x \sim \mu_V}[f_x]\right)^2\right].$$

*In matrix notation, define the matrix $M \in \mathbb{R}^{[k] \times \mathcal{X}}$ by $m_{v,x} = (\lambda_v(x) - \mu_v(x))$. Then*

$$\mathbb{E}_{V \sim \pi}\left[D_{KL}(\mathcal{T}_{\mathcal{M}}(\lambda_V^n) \| \mathcal{T}_{\mathcal{M}}(\mu_V^n))\right] \leq O(n\varepsilon^2) \cdot \|M\|_{\ell_\infty \to L_2(\pi)}^2.$$

Note that the statement of Lemma 8 is slightly different than the statement given in Edmonds et al. [2019], but the result as stated here is an immediate consequence of the original proof.

Our lower bound against agnostic learning will construct families $\{\lambda_1, \ldots, \lambda_k\}$ and $\{\mu_1, \ldots, \mu_k\}$ of distributions on $\mathcal{U} \times \{\pm 1\}$, as well as a distribution $\pi$ over $[k]$. The idea will be to construct these distributions so that, when $\mathcal{M}$ is an agnostic learner for $\mathcal{C}$, then, for any fixed $v \in [k]$,

$$D_{KL}(\mathcal{T}_{\mathcal{M}}(\lambda_v^n) \| \mathcal{T}_{\mathcal{M}}(\mu_v^n)) \geq 2d_{TV}(\mathcal{T}_{\mathcal{M}}(\lambda_v^n) \| \mathcal{T}_{\mathcal{M}}(\mu_v^n))^2 \geq \Omega(1),$$

where the first inequality is just Pinsker's inequality, and the second one will follow from our construction of $\lambda_v$ and $\mu_v$. Then the lower bound will follow by Lemma 8, as long as we can make sure that $\|M\|_{\ell_\infty \to L_2(\pi)}$ is small.

## 2.3 Duality and Hard Distributions

For the construction of hard families of distributions, it will be convenient to make use of the following dual formulation, shown in Edmonds et al. [2019].

**Lemma 9.** *Let $D \in \mathbb{R}^{\mathcal{C}^2 \times \mathcal{U}}$ be the difference matrix of a concept class $\mathcal{C}$, as given by (1). Then,*

$$\gamma_2(D, \alpha) = \max\left\{\frac{D \bullet U - \alpha\|U\|_1}{\gamma_2^*(U)} \; : \; U \in \mathcal{C}^2 \times \mathcal{U}, \; U \neq 0\right\}. \tag{2}$$

For an arbitrary concept class $\mathcal{C} \subseteq \{\pm 1\}^{\mathcal{U}}$, let $U \in \mathbb{R}^{\mathcal{C}^2 \times \mathcal{U}}$ witness (2), so that

$$\gamma_2(D, \alpha) = \frac{D \bullet U - \alpha\|U\|_1}{\gamma_2^*(U)}. \tag{3}$$

By normalizing $U$, we may assume, without loss of generality, that $\|U\|_1 = 1$. Moreover, we can assume that, for any $c, c' \in \mathcal{C}$, $\sum_{a \in \mathcal{U}} d_{(c,c'),a} u_{(c,c'),a} \geq 0$. Otherwise, $U$ cannot achieve (2), since we can multiply the row of $U$ indexed by $(c, c')$ by $-1$, which increases $D \bullet U$ and does not change $\|U\|_1$ or $\gamma_2^*(U)$.

We will consider the matrices $U^+, U^- \in \mathbb{R}^{\mathcal{C}^2 \times \mathcal{U}}$ with non-negative entries which satisfy $U = U^+ - U^-$, so that $U^+$ and $U^-$ correspond to the positive and negative entries of $U$ respectively. We define the distribution $\pi$ on $\mathcal{C}^2$ by

$$\pi(c, c') = \sum_{a \in \mathcal{U}} u_{(c,c'),a}. \tag{4}$$

Then, for $c, c' \in \mathcal{C}$, consider the distribution $\lambda_{c,c'}$ on $\mathcal{U} \times \{\pm 1\}$ given by

$$\lambda_{c,c'}(a, 1) = \frac{u^+_{(c,c'),a}}{\pi(c, c')}, \qquad \lambda_{c,c'}(a, -1) = \frac{u^-_{(c,c'),a}}{\pi(c, c')} \tag{5}$$

Similarly, let $\mu_{c,c'}$ be the distribution on $\mathcal{U} \times \{\pm 1\}$ given by

$$\mu_{c,c'}(a, 1) = \frac{u^-_{(c,c'),a}}{\pi(c, c')}, \qquad \mu_{c,c'}(a, -1) = \frac{u^+_{(c,c'),a}}{\pi(c, c')}. \tag{6}$$

Since $U$ has unit $\ell_1$ norm, the above distributions are well-defined. Note that $\lambda_{c,c'}$ and $\mu_{c,c'}$ have the same marginal on $\mathcal{U}$ which we denote $\kappa_{(c,c')}$. In particular, $\kappa_{(c,c')}(a) = \frac{|u_{(c,c'),a}|}{\pi(c,c')}$. Meanwhile, $\lambda_{c,c'}$ always gives $a$ the label $b = \text{sign}(u_{(c,c'),a})$, while $\mu_{c,c'}$ always gives $a$ the label $b = -\text{sign}(u_{(c,c'),a})$. It will be useful to have notation for one of these labelling functions, so define $s_{c,c'} : \mathcal{U} \to \{\pm 1\}$ by $s_{c,c'}(a) = \text{sign}(u_{(c,c'),a})$.

Consider the following relationship between $U$ and the distributions we have constructed.

$$u_{(c,c'),a} = \pi(c, c') \left( \lambda_{c,c'}(a, 1) - \mu_{c,c'}(a, 1) \right) = \pi(c, c') \kappa_{c,c'}(a) s_{c,c'}(a)$$

Note that

$$\sum_{a \in \mathcal{U}} d_{(c,c'),a} u_{(c,c'),a} = \pi(c, c') \cdot \sum_{a \in \mathcal{U}} \frac{1}{2} \cdot \kappa_{c,c'}(a) \cdot [c(a) s_{c,c'}(a) - c'(a) s_{c,c'}(a)]$$

$$= \pi(c, c') \cdot \left( L_{\lambda_{c,c'}}(c) - L_{\lambda_{c,c'}}(c') \right). \tag{7}$$

Similarly,

$$\sum_{a \in \mathcal{U}} d_{(c,c'),a} u_{(c,c'),a} = \pi(c, c') \cdot \left( L_{\mu_{c,c'}}(c') - L_{\mu_{c,c'}}(c) \right). \tag{8}$$

Hence,

$$D \bullet U = \mathop{\mathbb{E}}_{(c,c') \sim \pi} \left[ \left( L_{\lambda_{c,c'}}(c') - L_{\lambda_{c,c'}}(c) \right) \right] = \mathop{\mathbb{E}}_{(c,c') \sim \pi} \left[ \left( L_{\mu_{c,c'}}(c) - L_{\mu_{c,c'}}(c') \right) \right].$$

Whenever $\mathcal{C}$ contains at least two distinct concepts, $\gamma_2(D, \alpha) > 0$, and then (3) implies $D \bullet U > \alpha$. By the equations above, this implies that, on average with respect to $(c, c') \sim \pi$, the loss of $c$ is greater by $\alpha$ than the loss of $c'$ on $\lambda_{c,c'}$. Likewise, on average, the loss of $c'$ is greater by $\alpha$ than the loss of $c$ on $\mu_{c,c'}$. We will see later that, if we can obtain these properties in the worst case over all $(c, c')$, rather than only on average, then no hypothesis can fit both $\lambda_{c,c'}$ and $\mu_{c,c'}$ for any $c, c' \in \mathcal{C}$. The following section modifies the distributions we have constructed so as to obtain the required properties.

## 2.4 Lower bound derivation

The following lemma gives a modification of the hard distributions we constructed, giving us the properties we need for our lower bound. The proof is deferred to Appendix C.

**Lemma 10.** *Let $\mathcal{C}$ be a concept class. Let $D$ be the matrix given by (1). Let $U \in \mathbb{R}^{\mathcal{C}^2 \times \mathcal{U}}$, $\|U\|_1 = 1$, satisfy (3). Then there exist probability distributions $\widetilde{\lambda}_{c,c'}$ and $\widetilde{\mu}_{c,c'}$ over $\mathcal{U} \times \{\pm 1\}$, and a distribution $\widetilde{\pi}$ over $\mathcal{C}^2$ such that:*

1. *For all $(c, c')$ in the support of $\widetilde{\pi}$, $L_{\widetilde{\lambda}_{c,c'}}(c) - L_{\widetilde{\lambda}_{c,c'}}(c') \geq \frac{\alpha}{O(\log(1/\alpha))}$.*

2. *For all $(c, c')$ in the support of $\widetilde{\pi}$, $L_{\widetilde{\mu}_{c,c'}}(c') - L_{\widetilde{\mu}_{c,c'}}(c) \geq \frac{\alpha}{O(\log(1/\alpha))}$.*

3. *The matrix $\widetilde{U} \in \mathbb{R}^{\mathcal{C}^2 \times \mathcal{U}}$ with entries $\widetilde{u}_{(c,c'),a} = \widetilde{\pi}(c,c')(\widetilde{\lambda}_{c,c'}(a) - \widetilde{\mu}_{c,c'}(a))$ satisfies $\gamma_2^*(\widetilde{U}) \leq \frac{\alpha \gamma_2^*(U)}{D \bullet U}$.*

We also want to bound the operator norm, which appears in Lemma 8, in terms of $\gamma_2^*(U)$. To do so, we use the following lemma from Edmonds et al. [2019].

**Lemma 11** ([Edmonds et al., 2019]). *Let $U$ and $M$ be $k \times T$ matrices, and let $\pi$ be a probability distribution on $[k]$ such that, for any $i \in [k], j \in [T]$, we have $u_{i,j} = \pi(i)m_{i,j}$. Then there exists a probability distribution $\widehat{\pi}$ on $[k]$, with support contained in the support of $\pi$, such that $\|M\|_{\ell_\infty \to L_2(\widehat{\pi})} \leq 4\gamma_2^*(U)$.*

Recall that we also want to obtain a lower bound on $d_{\mathrm{TV}}(\mathcal{T}_{\mathcal{M}}(\lambda_{c,c'}^n)\|\mathcal{T}_{\mathcal{M}}(\mu_{c,c'}^n))$ when $\mathcal{M}$ is a learning algorithm for $\mathcal{C}$. For this purpose, we apply the following lemma, whose proof is also deferred to Appendix C. The main observation in the proof is, for any hypothesis $h : \mathcal{C} \to \{\pm 1\}$, and any distributions $\lambda$ and $\mu$ satisfying the conditions of the lemma, we have $L_\lambda(h) + L_\mu(h) = 1$.

**Lemma 12.** *Let $\lambda$ and $\mu$ be distributions on $\mathcal{U} \times \{\pm 1\}$. Assume that $\lambda$ and $\mu$ have the same marginal on $\mathcal{U}$. Also assume that $\lambda$ is labelled by some $s : \mathcal{U} \to \{\pm 1\}$ while $\mu$ is labelled by $-s$. Finally, assume that for some $c, c' \in \mathcal{C}$, $L_\mu(c') - L_\mu(c) > \alpha$. If $h : \mathcal{U} \to \{\pm 1\}$ satisfies $L_\lambda(h) \leq L_\lambda(c') + \alpha/4$, then $L_\mu(h) > L_\mu(c) + 3\alpha/4$. Hence, if $\mathcal{M}$ is an algorithm which $(\alpha/4, \beta)$-learns $\mathcal{C}$ from $n$ samples, then $d_{\mathrm{TV}}(\mathcal{M}(\lambda^n), \mathcal{M}(\mu^n)) \geq 1 - 2\beta$.*

Finally, with these results at our disposal, we may obtain the lower bound of Lemma 7.

*Proof of Lemma 7.* Let $U \in \mathbb{R}^{\mathcal{C}^2 \times \mathcal{U}}$, $\|U\|_1 = 1$, satisfy (3). Let $\widetilde{\pi}$, together with $\widetilde{\lambda}_{c,c'}$ and $\widetilde{\mu}_{c,c'}$ be the distributions guaranteed to exist by Lemma 10 and let $\widetilde{U} \in \mathbb{R}^{\mathcal{C}^2 \times \mathcal{U}}$ be the corresponding matrix with entries

$$\widetilde{u}_{(c,c'),a} = \widetilde{\pi}(c,c')\left(\widetilde{\lambda}_{c,c'}(a,1) - \widetilde{\mu}_{c,c'}(a,1)\right) = -\widetilde{\pi}(c,c')\left(\widetilde{\lambda}_{c,c'}(a,-1) - \widetilde{\mu}_{c,c'}(a,-1)\right)$$

Let $M$ be the matrix with entries $m_{(c,c'),a} = \widetilde{u}_{(c,c'),a}/\widetilde{\pi}(c,c')$. By Lemma 11, there exists some distribution $\hat{\pi}$ with support contained in that of $\widetilde{\pi}$ such that

$$\|M\|_{\ell_\infty \to L_2(\widehat{\pi})} \leq 4\gamma_2^*(\widetilde{U}) \leq \frac{4\alpha\gamma_2^*(U)}{D \bullet U},$$

where the last inequality follows from Lemma 10. Combining Lemma 8 with the dual formulation (3) gives

$$\mathbb{E}_{(c,c')\sim\widehat{\pi}}\left[D_{\mathrm{KL}}(\mathcal{T}_{\mathcal{M}}(\widetilde{\lambda}_{c,c'}^n)\|\mathcal{T}_{\mathcal{M}}(\widetilde{\mu}_{c,c'}^n))\right] \leq O(n\varepsilon^2) \cdot \|M\|_{\ell_\infty \to L_2(\widehat{\pi})}^2$$

$$\leq O(n\varepsilon^2) \cdot \left(\frac{\alpha\gamma_2^*(U)}{D \bullet U}\right)^2 = O(n\varepsilon^2) \cdot \left(\frac{\alpha}{\gamma_2(D,\alpha)}\right)^2.$$

Now let

$$\alpha' = \frac{1}{4}\left(\min_{c,c'} L_{\mu_{c,c'}}(c') - L_{\mu_{c,c'}}(c)\right) \geq \frac{\alpha}{O(\log(1/\alpha))},$$

where the last inequality is by Lemma 10. By Lemma 12, if $\mathcal{M}$ $(\alpha', \beta)$-learns $\mathcal{C}$ for some $\beta = \frac{1}{2} - \Omega(1)$, then $\mathbb{E}_{(c,c')\sim\widetilde{\pi}}\left[D_{\mathrm{KL}}(\mathcal{T}_{\mathcal{M}}(\widetilde{\lambda}_{c,c'}^n)\|\mathcal{T}_{\mathcal{M}}(\widetilde{\mu}_{c,c'}^n))\right] = \Omega(1)$. This implies $n = \Omega\left(\left(\frac{\gamma_2(D,\alpha)}{\varepsilon\alpha}\right)^2\right)$, as was to be proved. $\square$

## 3 Refutation versus Learning: Realizable Case

In this section, we present our algorithm for realizable learning and refutation for non-interactive LDP. For a concept class $\mathcal{C} : \mathcal{U} \to \{\pm 1\}$, we define a quantity $\eta(\mathcal{C}, \alpha)$ and argue that it gives an upper bound on the sample complexity for realizable learning of $\mathcal{C}$.

**Definition 13.** *Let $\mathcal{C} : \mathcal{U} \to \{\pm 1\}$ be a concept class. Let*

$$K_{\mathcal{C}} = \left\{W \in \mathbb{R}^{\mathcal{C} \times (\mathcal{U} \times \{\pm 1\})} \ : \ |w_{c,(a,c(a))}| \leq \alpha \text{ and } w_{c,(a,-c(a))} \geq 1 \ \forall c \in \mathcal{C}, a \in \mathcal{U}\right\}. \quad (9)$$

*Let*

$$K'_{\mathcal{C}} = \left\{ \widetilde{W} \in \mathbb{R}^{\mathcal{C} \times (\mathcal{U} \times \{\pm 1\})} \; : \; \exists W \in K_{\mathcal{C}}, \; \exists \theta \in \mathbb{R}^{\mathcal{C}}, \; \widetilde{W} = W + \theta \mathbf{1}^T \right\}, \qquad (10)$$

*where $\mathbf{1}^T$ is the all-ones row vector indexed over $\mathcal{C}$, so that $\widetilde{W} = W + \theta \mathbf{1}^T$ is the matrix obtained by shifting each row $c$ of $W$ in each entry by $\theta_c$.*

*Then define*

$$\eta(\mathcal{C}, \alpha) = \min \left\{ \gamma_2(\widetilde{W}) : \widetilde{W} \in K'_{\mathcal{C}} \right\}.$$

The idea is that each row of $W$ defines a statistical query corresponding to a concept, $q_c(a, b) = w_{c,(a,b)}$. The statistical query corresponding to the true concept that was used to label the data will have a small value, whereas any query corresponding to a concept with large loss will have a large value. The next theorem formalizes this argument.

**Theorem 14.** *Let $\mathcal{C} \subseteq \{\pm 1\}^{\mathcal{U}}$ be a concept class. Let $\varepsilon > 0$, $\alpha, \beta \in (0, 1]$. Then there exists an $\varepsilon$-LDP mechanism which may be used to both $(3\alpha, \beta)$-learn $\mathcal{C}$ realizably and $(3\alpha, \beta)$-refute $\mathcal{C}$ realizably with $n$ samples, where*

$$n = O\left( \frac{\eta(\mathcal{C}, \alpha) \cdot \log(|\mathcal{C}|/\beta)}{\varepsilon^2 \alpha^2} \right).$$

*Proof.* As per Definition 13, let $\widetilde{W} \in K'_{\mathcal{C}}$ be the matrix that witnesses $\eta(\mathcal{C}, \alpha)$ and let $W \in K_{\mathcal{C}}$ and $\theta \in \mathbb{R}^{\mathcal{C}}$ be the matrix and vector which witness $\widetilde{W} \in K'_{\mathcal{C}}$. If we can answer the statistical queries given by $\widetilde{W}$, then we can answer the queries given by $W$ with the same accuracy by subtracting $\theta_c$ from the answer to the query for concept $c$.

By the definition of $W$, if, for some $c \in \mathcal{C}$, $\lambda$ is supported on on those $(a, b) \in \mathcal{U} \times \{\pm 1\}$ which satisfy $c(a) = b$, then the value of the query corresponding to $c$ is bounded as

$$\mathbb{E}_{(a,b) \sim \lambda} \left[ w_{c,(a,b)} \right] = \mathbb{E}_{(a,b) \sim \lambda} \left[ w_{c,(a,c(a))} \right] \leq \alpha.$$

Meanwhile, for an arbitrary distribution $\lambda$ on $\mathcal{U} \times \{\pm 1\}$, the value of the query corresponding to $c \in \mathcal{C}$ may be bounded as

$$\mathbb{E}_{(a,b) \sim \lambda} \left[ w_{c,(a,b)} \right] \geq \mathbb{P}_{(a,b) \sim \lambda} \left[ b \neq c(a) \right] - \alpha \cdot \mathbb{P}_{(a,b) \sim \lambda} \left[ b = c(a) \right] \geq L_\lambda(c) - \alpha.$$

In particular, if $L_\lambda(c) \geq 3\alpha$, then $\mathbb{E}_{(a,b) \sim \lambda} \left[ w_{c,(a,b)} \right] \geq 2\alpha$.

It follows that, by approximating the statistical queries given by $W$ with worst-case error $\frac{\alpha}{4}$, we can distinguish the case where $\lambda$ agrees with some $c \in \mathcal{C}$ from the case where, for all concepts $c \in \mathcal{C}$, $L_\lambda(c) \geq 3\alpha$. In the former case, returning some $c' \in \mathcal{C}$ where our estimate of $\mathbb{E}_{(a,b) \sim \lambda} \left[ w_{c,(a,b)} \right]$ is strictly less than $2\alpha$ guarantees $L_\lambda(c) < 3\alpha$.

To complete the proof, it suffices to apply the upper bound from Edmonds et al. [2019] which says that, to answer the collection of statistical queries given by $\widetilde{W}$ under non-interactive $\varepsilon$-LDP, with accuracy $\alpha/4$ and probability of failure at most $\beta$, the number of samples required is at most

$$O\left( \frac{\gamma_2(\widetilde{W}) \log(|\mathcal{C}|/\beta)}{\varepsilon^2 \alpha^2} \right) = O\left( \frac{\eta(\mathcal{C}, \alpha) \log(|\mathcal{C}|/\beta)}{\varepsilon^2 \alpha^2} \right). \qquad \square$$

## 3.1 Lower bound

Our lower bound will follow a similar strategy as in the agnostic case. However, our construction of hard distributions will be tailored to $\eta(\mathcal{C}, \alpha)$ and its dual.

### 3.1.1 Duality

We will again use convex duality in our lower bound. We will express $\eta(\mathcal{C}, \alpha)$ as a maximum over dual matrices $U$, and we will use an optimal $U$ to construct 'hard distributions' for realizable refutation. To this end, consider the following duality lemma, proved in Appendix D.

**Lemma 15.** *For any concept class $\mathcal{C} \subseteq \{\pm 1\}^{\mathcal{U}}$ and any $\alpha$,*

$$\eta(\mathcal{C}, \alpha) = \max_{U \in S_{\mathcal{C}}} \frac{\sum_{c \in \mathcal{C}, a \in \mathcal{U}} (u_{c,(a,-c(a))} - \alpha |u_{c,(a,c(a))}|)}{\gamma_2^*(U)}, \tag{11}$$

*where we define*

$$S_{\mathcal{C}} := \left\{ U \in \mathbb{R}^{\mathcal{C} \times (\mathcal{U} \times \{\pm 1\})} : \forall c \in \mathcal{C}, \sum_{a \in \mathcal{U}} (u_{c,(a,c(a))} + u_{c,(a,-c(a))}) = 0 \right.$$

$$\left. \text{and, } \forall c \in \mathcal{C}, \forall a \in \mathcal{U}, u_{c,(a,-c(a))} \geq 0 \right\}.$$

### 3.1.2 Hard distributions

Let $U \in \mathbb{R}^{\mathcal{C} \times (\mathcal{U} \times \{\pm 1\})}$ witness (11). By normalizing, we may assume without loss of generality that $\|U\|_1 = 1$. We will consider the matrices $U^+, U^- \in \mathbb{R}^{m \times N}$ with non-negative entries which satisfy $U = U^+ - U^-$ so that $U^+$ and $U^-$ correspond to the positive and negative entries of $U$ respectively. We define the distribution $\pi$ on $\mathcal{C}$ given by $\pi(c) = \sum_{(a,b) \in \mathcal{U} \times \{\pm 1\}} u_{c,(a,b)}$. Then, for each $c \in \mathcal{C}$, let $\lambda_c$ and $\mu_c$ be the distributions on $\mathcal{U} \times \{\pm 1\}$ given by

$$\lambda_c(a, b) = \frac{2u_{c,(a,b)}^+}{\pi(c)} \qquad \text{and} \qquad \mu_c(a, b) = \frac{2u_{c,a,b}^-}{\pi(c)}.$$

Since the rows of $U$ each sum to zero and have unit $\ell_1$ norm, the distributions $\lambda_c$ and $\mu_c$ are well-defined. Moreover, since $u_{c,(a,-c(a))} \geq 0$ for all $c \in \mathcal{C}, a \in \mathcal{U}$, the only negative entries of $U$ are those of the form $u_{c,(x,c(x))}$. This implies that the distribution $\mu_c$ always labels samples $a \in \mathcal{U}$ by $c(a)$.

### 3.1.3 Warm-up: single-concept case

Consider the case where $\mathcal{C}$ consists of a single concept $c$. Since $\eta(\mathcal{C}, \alpha) > 0$, then (11) implies

$$\sum_{a \in \mathcal{U}} u_{c,(a,-c(a))} > \sum_{a \in \mathcal{U}} \alpha |u_{c,(a,c(a))}|. \tag{12}$$

Hence,

$$\mathbb{P}_{(a,b) \sim \lambda_c}[c(a) \neq b] - \mathbb{P}_{(a,b) \sim \mu_c}[c(a) \neq b] > \alpha \cdot \left( \mathbb{P}_{(a,b) \sim \lambda_c}[c(a) = b] + \mathbb{P}_{(a,b) \sim \mu_c}[c(a) = b] \right).$$

Using

$$\mathbb{P}_{(a,b) \sim \mu_c}[c(a) = b] = 1 \tag{13}$$

and rearranging, this gives

$$L_{\lambda_c}(c) = \mathbb{P}_{(a,b) \sim \lambda_c}[c(a) \neq b] > \frac{2\alpha}{1 + \alpha}.$$

In other words, if we can distinguish a distribution on $\mathcal{U} \times \{\pm 1\}$ which labels samples according to $c$ from one which disagrees with $c$ with probability greater than $\frac{2\alpha}{1+\alpha}$, then we can distinguish between $\lambda_c$ and $\mu_c$.

### 3.1.4 General case

Appendix D.2 generalizes the lower bound of the previous section to the general case where the concept class is not restricted to a single concept.

The first issue which needs to be addressed in the general case is that (12), rather than holding in worst case over all concepts, holds on average for a concept $c \in \mathcal{C}$ drawn from the distribution $\pi$. This issue is handled by applying the binning result of Lemma 19.

The second issue which needs to be addressed is that, while each $c \in \mathcal{C}$ is guaranteed not to fit the corresponding distribution $\lambda_c$, as with (13), it may hold that some other $h : \mathcal{U} \to \{\pm 1\}$ has small loss on $\lambda_c$. This is remedied by mixing a distribution $\sigma_c$ which agrees with $c$ into the distribution $\lambda_c$. This guarantees that every $h : \mathcal{U} \to \{\pm 1\}$ has large loss on $\sigma_c$.

Altogether, we obtain the lower bound of Theorem 1.

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
