# A    Preliminaries

## A.1    Norms

For a set $\mathcal{S}$, the $\ell_1$, $\ell_2$ and $\ell_\infty$ norms on $\mathbb{R}^{\mathcal{S}}$ are given respectively by

$$\|a\|_1 = \sum_{v \in \mathcal{S}} |a_v|, \quad \|a\|_2 = \sqrt{\sum_{v \in \mathcal{S}} (a_v)^2}, \quad \|a\|_\infty = \max_{v \in \mathcal{S}} |a_v|.$$

Given a probability distribution $\pi$ on $\mathcal{S}$, we consider the norms $\|\cdot\|_{L_1(\pi)}$ and $\|\cdot\|_{L_2(\pi)}$ on $\mathbb{R}^{\mathcal{S}}$, given by

$$\|a\|_{L_1(\pi)} = \sum_{v \in \mathcal{S}} \pi(v) |a_v|, \quad \|a\|_{L_2(\pi)} = \sqrt{\sum_{v \in \mathcal{S}} \pi(v)(a_v)^2}.$$

We also take advantage of a number of matrix norms. For norms $\|\cdot\|_\zeta$ and $\|\cdot\|_\xi$ on $\mathbb{R}^{\mathcal{S}'}$ and $\mathbb{R}^{\mathcal{S}}$ respectively, we consider the *matrix operator norm* of $M \in \mathbb{R}^{\mathcal{S} \times \mathcal{S}'}$ given by

$$\|M\|_{\zeta \to \xi} = \max_{a \in \mathbb{R}^{\mathcal{S}} \setminus \{0\}} \frac{\|Ma\|_\xi}{\|a\|_\zeta}.$$

For the special case of $\|M\|_{\ell_s \to \ell_t}$, we will simply write $\|M\|_{s \to t}$. Of particular importance are $\|M\|_{1 \to \infty}$ which corresponds to the largest entry of $M$, $\|M\|_{1 \to 2}$, which corresponds to the maximum $\ell_2$ norm of a column of $M$, and $\|M\|_{2 \to \infty}$, which corresponds to the maximum $\ell_2$ norm of a row of $M$.

The *inner product* of two matrices $M$ and $N$ in $\mathbb{R}^{\mathcal{S} \times \mathcal{S}'}$ is defined by $M \bullet N = \mathrm{Tr}(M^\top N) = \sum_{u \in \mathcal{S}, v \in \mathcal{S}'} m_{u,v} n_{u,v}$.

The *factorization norm* known as the $\gamma_2$ norm is given for $M \in \mathbb{R}^{\mathcal{S} \times \mathcal{S}'}$ by

$$\gamma_2(M) = \min\{\|R\|_{2 \to \infty} \|A\|_{1 \to 2} : RA = M\}.$$

The $\gamma_2$ norm is, indeed, a norm, i.e., it is non-negative, $\gamma_2(M) = 0$ if and only if $M = 0$, for any real $s$ we have $\gamma_2(sM) = |s|\gamma_2(M)$, and we also have the triangle inequality $\gamma_2(M + N) \leq \gamma_2(M) + \gamma_2(N)$.

The approximate $\gamma_2$ norm is the smallest $\gamma_2$ norm of a matrix that approximates the given matrix entrywise up to an additive $\alpha$, i.e.,

$$\gamma_2(M, \alpha) = \min\{\gamma_2(\widetilde{M}) : \|\widetilde{M} - M\|_{1 \to \infty} \leq \alpha\}.$$

The dual $\gamma_2$ norm of a matrix $G$ in $\mathbb{R}^{\mathcal{S} \times \mathcal{S}'}$ is given by

$$\gamma_2^*(k) = \max\{M \bullet N : \gamma_2(M) \leq 1\} = \max_{f,g} \sum_{u \in \mathcal{S}, v \in \mathcal{S}'} n_{u,v} f(u) g(v),$$

where the second max ranges over functions $f : \mathcal{S} \to B_2$ and $g : \mathcal{S}' \to B_2$ that map the index sets of the rows and columns of $N$, respectively, to vectors of $\ell_2$ norm at most 1.

## A.2    Differential privacy

Let $\mathcal{X}$ denote the *data universe*. A generic element from $\mathcal{X}$ will be denoted by $x$. We consider *datasets* of the form $X = (x_1, \ldots, x_n) \in \mathcal{X}^n$, each of which is identified with its *histogram* $h \in \mathbb{Z}_{\geq 0}^{\mathcal{X}}$ where, for every $x \in \mathcal{X}$, $h_x = |\{i : x_i = x\}|$, so that $\|h\|_1 = n$. To refer to a dataset, we use $X$ and $h$ interchangeably. A pair of datasets $X = (x_1, \ldots, x_i, \ldots, x_n)$ and $X' = (x_1, \ldots, x_i', \ldots, x_n)$ are called *adjacent* if $X'$ is obtained from $X$ by replacing an element $x_i$ of $X$ with a new universe element $x_i'$.

For parameters $\varepsilon > 0$, an $\varepsilon$-*differentially private* ($\varepsilon$-DP) mechanism [Dwork et al., 2006] is a randomized function $\mathcal{M} : \mathcal{X}^n \to \mathcal{Z}$ which, for all adjacent datasets $X$ and $X'$, for all outcomes $S \subseteq \mathcal{Z}$, satisfies

$$\Pr_{\mathcal{M}}[\mathcal{M}(X) \in S] \leq e^\varepsilon \Pr_{\mathcal{M}}[\mathcal{M}(X') \in S].$$

Of special interest are $\varepsilon$-differentially private mechanisms $\mathcal{M}_i : \mathcal{X} \to \mathcal{Y}$ which take a singleton dataset $X = \{x\}$ as input. These are referred to as *local randomizers*. A sequence of $\varepsilon$-differentially private local randomizers $\mathcal{M}_1, \ldots, \mathcal{M}_n$, together with a *post-processing function* $\mathcal{A} : \mathcal{Y}^n \to \mathcal{Z}$, specify a *(non-interactive) locally $\varepsilon$-differentially private ($\varepsilon$-LDP) mechanism* $\mathcal{M} : \mathcal{X}^n \to \mathcal{Z}$ [Evfimievski et al., 2003, Dwork et al., 2006, Kasiviswanathan et al., 2008]. When the local mechanism $\mathcal{M}$ is applied to a dataset $X$, we refer to $\mathcal{T}_\mathcal{M}(X) = (\mathcal{M}_1(x_1), \ldots, \mathcal{M}_n(x_n))$ as the *transcript* of the mechanism. Then the output of the mechanism is given by $\mathcal{M}(X) = \mathcal{A}(\mathcal{T}_\mathcal{M}(X))$.

A *linear query* is specified by a bounded function $q : \mathcal{X} \to \mathbb{R}$. Abusing notation slightly, its answer on a dataset $X$ is given by $q(X) = \frac{1}{n} \sum_{i=1}^n q(x_i)$. We also extend this notation to distributions: if $\lambda$ is a distribution on $\mathcal{X}$, then we write $q(\lambda)$ for $\mathbb{E}_{x \sim \lambda}[q(x)]$. A *workload* is a set of linear queries $Q = \{q_1, \ldots, q_k\}$, and $Q(X) = (q_1(X), \ldots, q_k(X))$ is used to denote their answers. The answers on a distribution $\lambda$ on $\mathcal{X}$ are denoted by $Q(\lambda) = (q_1(\lambda), \ldots, q_k(\lambda))$. We will often represent $Q$ by its *workload matrix* $W \in \mathbb{R}^{Q \times \mathcal{X}}$ with entries $w_{q,X} = q(X)$. In this notation, the answers to the queries are given by $\frac{1}{n} W h$. We will often use $Q$ and $W$ interchangeably.

## A.3 PAC learning

A concept $c : \mathcal{U} \to \{\pm 1\}$ from a concept class $\mathcal{C}$ identifies each sample $a \in \mathcal{U}$ with a label $c(a)$. The *empirical loss* of the concept $c$ on a dataset $X = ((a_1, b_1), \ldots, (a_n, b_n)) \in (\mathcal{U} \times \{\pm 1\})^n$, denoted $L_X(c)$, is given by

$$L_X(c) = \frac{1}{n} \sum_{i=1}^n (\mathbb{I}[c(a_i) \neq b_i])$$

For a distribution $\lambda$ on $\mathcal{U} \times \{\pm 1\}$, the *population loss* of $c$ on $\lambda$, denoted $L_\lambda(c)$ is given by $L_\lambda(c) = \mathbb{P}_{(a,b) \sim \lambda}[c(a) \neq b]$.

We will say that a mechanism $\mathcal{M} : (\mathcal{U} \times \{\pm 1\})^n \to \{\pm 1\}^\mathcal{U}$ *$(\alpha, \beta)$-learns $\mathcal{C}$ agnostically* with $n$ samples if, for any distribution $\lambda$ over $\mathcal{U} \times \{\pm 1\}$, given as input a random dataset $X$ drawn i.i.d. from $\lambda$, the mechanism returns some *hypothesis* $h \in \mathcal{U} \to \{\pm 1\}$ which satisfies

$$\mathbb{P}_{X, \mathcal{M}} \left[ L_\lambda(h) \leq \min_{c \in \mathcal{C}} L_\lambda(c) + \alpha \right] \geq 1 - \beta.$$

Realizable learning is an important special case of agnostic learning where the underlying distribution agrees with some concept. We say that $\mathcal{M} : (\mathcal{U} \times \{\pm 1\})^n \to \{\pm 1\}^\mathcal{U}$ *$(\alpha, \beta)$-learns $\mathcal{C}$ realizably* with $n$ samples if, whenever $\lambda$ is a distribution over $\mathcal{U} \times \{\pm 1\}$ which satisfies $\mathbb{P}_{(a,b) \sim \lambda}[c(a) = b] = 1$ for some unknown $c \in \mathcal{C}$, then, given a random dataset $X$ drawn i.i.d. from $\lambda$, the mechanism returns a *hypothesis* $h \in \mathcal{U} \to \{\pm 1\}$ which satisfies

$$\mathbb{P}_{X, \mathcal{M}}[L_\lambda(h) \leq \alpha] \geq 1 - \beta.$$

The problem of refutation asks whether the underlying distribution is well approximated by the concept class. In particular, for $\theta \in [0, 1]$, we will say that $\mathcal{M}_\theta : (\mathcal{U} \times \{\pm 1\})^n \to \{\pm 1\}$ *$(\alpha, \beta)$-refutes $\mathcal{C}$ for threshold $\theta$* if the following two conditions are met:

1. When $\lambda$ is a distribution on $\mathcal{U} \times \{\pm 1\}$ which satisfies $L_\lambda(c) \leq \theta$ for some $c \in \mathcal{C}$,

$$\mathbb{P}_{X, \mathcal{M}}[\mathcal{M}(X) = 1] \geq 1 - \beta;$$

2. When $\lambda$ is a distribution on $\mathcal{U} \times \{\pm 1\}$ which, for all $h \in \{\pm 1\}^\mathcal{U}$, satisfies $L_\lambda(h) > \theta + \alpha$, then

$$\mathbb{P}_{X, \mathcal{M}}[\mathcal{M}(X) = -1] \geq 1 - \beta.$$

Realizable refutation is a special case of agnostic refutation where the goal is to recognize whether the underlying distribution is labeled by a concept from the concept class. We say that $\mathcal{M}_\theta : (\mathcal{U} \times \{\pm 1\})^n \to \{\pm 1\}$ *$(\alpha, \beta)$-refutes $\mathcal{C}$ realizably* if it $(\alpha, \beta)$-refutes $\mathcal{C}$ for threshold 0.

# B  Equivalence of approximate $\gamma_2$ norms of difference and concept matrices

We prove Lemma 6, the equivalence of the approximate $\gamma_2$ norm for matrices $W$ and $D$, via the following three Lemmas.

**Lemma 16.** *Let $\mathcal{C}$ be a concept class with concept matrix $W \in \mathbb{R}^{\mathcal{C} \times \mathcal{U}}$ and difference matrix $D \in \mathbb{R}^{\mathcal{C}^2 \times \mathcal{U}}$. Then $\gamma_2(D, \alpha) \leq \gamma_2(W, \alpha)$.*

**Lemma 17.** *Let $\mathcal{C}$ be a concept class closed under negation. Let $W \in \mathcal{C} \times \mathcal{U}$ be its concept matrix (Definition 3) and let $D \in \mathbb{R}^{\mathcal{C}^2 \times \mathcal{U}}$ be its difference matrix (Definition 5). Then $\gamma_2(W, \alpha) \leq \gamma_2(D, \alpha)$.*

**Lemma 18.** *Let $\mathcal{C}$ be a concept class. Let $W \in \mathbb{R}^{\mathcal{C} \times \mathcal{U}}$ be its concept matrix (Definition 3) and let $D \in \mathbb{R}^{\mathcal{C}^2 \times \mathcal{U}}$ be its difference matrix (Definition 5). Then $\gamma_2(W, \alpha) \leq 2\gamma_2(D, \alpha/2) + 1$.*

*Proof of Lemma 16.* Let $\widetilde{W} \in \mathcal{C} \times \mathcal{U}$ witness $\gamma_2(W, \alpha)$ so that $\|W - \widetilde{W}\|_{1 \to \infty} \leq \alpha$ and $\gamma_2(W, \alpha) = \gamma_2(\widetilde{W})$.

Let $W' \in \mathbb{R}^{\mathcal{C}^2 \times \mathcal{U}}$ be the matrix with entries $w'_{(c,c'),a} = \widetilde{w}_{c,a}$. Similarly, let $W'' \in \mathbb{R}^{\mathcal{C}^2 \times \mathcal{U}}$ be the matrix with entries $w''_{(c,c'),a} = \widetilde{w}_{c',a}$. Since $W'$ and $W''$ are obtained from $\widetilde{W}$ by duplicating rows,

$$\gamma_2(W, \alpha) = \gamma_2(\widetilde{W}) = \gamma_2(W') = \gamma_2(W'').$$

Now consider the matrix $\widetilde{D} = \frac{1}{2}(W' - W'')$. By subadditivity and scaling properties,

$$\gamma_2(\widetilde{D}) \leq \frac{1}{2}(\gamma_2(W') + \gamma_2(W'')) = \gamma_2(\widetilde{W}).$$

Moreover, for all $c, c' \in \mathcal{C}$, $a \in \mathcal{U}$, the entry $\widetilde{d}_{(c,c'),a}$ of $\widetilde{D}$ approximates entry $d_{(c,c'),a}$ of $D$. Specifically,

$$\begin{aligned}
\left| \widetilde{d}_{(c,c'),a} - d_{(c,c'),a} \right| &= \left| \frac{\widetilde{w}_{c,a} - \widetilde{w}_{c',a}}{2} - \frac{c(a) - c'(a)}{2} \right| \\
&\leq \left| \frac{\widetilde{w}_{c,a} - c(a)}{2} \right| + \left| \frac{\widetilde{w}_{c',a} - c'(a)}{2} \right| \\
&\leq \alpha.
\end{aligned}$$

Hence $\|D - \widetilde{D}\|_{1 \to \infty} \leq \alpha$. Together with $\gamma_2(\widetilde{D}) \leq \gamma_2(\widetilde{W})$, this implies

$$\gamma_2(D, \alpha) \leq \gamma_2(\widetilde{D}) \leq \gamma_2(\widetilde{W}) = \gamma_2(W, \alpha). \qquad \square$$

*Proof of Lemma 17.* Row $c$ of $W$ is identical with row $(c, -c)$ of $D$. Hence, $W$ is obtained from $D$ by deleting some of its rows. Since the $\gamma_2$ norm is non-increasing under taking submatrices, it follows that $\gamma_2(W, \alpha) \leq \gamma_2(D, \alpha)$. $\qquad \square$

*Proof of Lemma 18.* Fix an arbitrary concept $c' \in \mathcal{C}$. Let $D'$ be the submatrix of $D$ which includes row $(c, c')$ of $D$ for each $c \in \mathcal{C}$. Then $D' = \frac{1}{2}\left(W - \mathbf{1}(c')^T\right)$ where $\mathbf{1}$ is the all-ones vector of dimension $|\mathcal{C}|$, and we identify $c'$ with a vector in $\mathbb{R}^{\mathcal{U}}$. Expressing our concept matrix as $W = 2D' + \mathbf{1}(c')^T$, we may apply the scaling and subadditivity properties of the approximate $\gamma_2$ norm to obtain

$$\begin{aligned}
\gamma_2(W, \alpha) &= \gamma_2(2D' + \mathbf{1}(c')^T, \alpha) \\
&\leq \gamma_2(2D', \alpha) + \gamma_2(\mathbf{1}(c')^T, 0) \\
&\leq 2\gamma_2(D', \alpha/2) + 1. \qquad \square
\end{aligned}$$

# C  Omitted Proofs for the Lower Bound on Agnostic Learning

In proving Lemma 10, we take advantage of the following technical lemma by a standard geometric binning argument.

**Lemma 19** ([Edmonds et al., 2019]). *Suppose that $a_1, \ldots, a_k \in [0,1]$ and that $\pi$ is a probability distribution over $[k]$. Then for any $\beta \in (0,1]$, there exists a set $S \subseteq [k]$ such that $\pi(S) \cdot \min_{v \in S} a_v \geq \frac{\sum_{v=1}^{k} \pi(v) a_v - \beta}{O(\log(1/\beta))}$.*

*Proof of Lemma 10.* Let $\pi$, together with $\lambda_{c,c'}$ and $\mu_{c,c'}$, be defined as in (4), (5) and (6). We will apply Lemma 19 to the values given, for $c, c' \in \mathcal{C}$, by

$$a_{c,c'} = L_{\lambda_{c,c'}}(c) - L_{\lambda_{c,c'}}(c) = L_{\mu_{c,c'}}(c) - L_{\mu_{c,c'}}(c).$$

Recall that we may assume, that for all $c, c' \in \mathcal{C}$ the following inequality holds

$$\sum_{a \in \mathcal{U}} d_{(c,c'),a} u_{(c,c'),a} \geq 0.$$

Together with (7), this gives $a_{c,c'} \geq 0$ for all $c, c' \in \mathcal{C}$.

By Lemma 19, there exists some $S \subset \mathcal{C}^2$ such that

$$\pi(S) \cdot \min_{(c,c') \in S} \left( L_{\lambda_{c,c'}}(c) - L_{\lambda_{c,c'}}(c') \right) \geq \frac{\mathbb{E}_{(c,c') \sim \pi} \left[ L_{\lambda_{c,c'}}(c) - L_{\lambda_{c,c'}}(c') \right] - \alpha/4}{O(\log(1/\alpha))} = \frac{D \bullet U - \alpha/4}{O(\log(1/\alpha))}.$$

By applying (8), we get the similar

$$\pi(S) \cdot \min_{(c,c') \in S} \left( L_{\mu_{c,c'}}(c') - L_{\mu_{c,c'}}(c) \right) \geq \frac{D \bullet U - \alpha/4}{O(\log(1/\alpha))}.$$

Let $\widetilde{\pi}$ be defined by

$$\widetilde{\pi}(c, c') = \begin{cases} \pi(c, c')/\pi(S), & \text{if } c, c' \in S \\ 0, & \text{otherwise.} \end{cases}$$

Let also $\tau = \frac{\alpha}{D \bullet U} \in (0, 1)$. For $(c, c') \in S$, let $\widetilde{\lambda}_{c,c'} = \lambda_{c,c'}$ and

$$\widetilde{\mu}_{c,c'} = (1 - \tau\pi(S))\lambda_{c,c'} + \tau\pi(S)\mu_{c,c'}.$$

It holds then, for $(c, c') \in S$, that

$$\widetilde{\lambda}_{c,c'} - \widetilde{\mu}_{c,c'} = \tau \cdot \pi(S) \cdot (\lambda_{c,c'} - \mu_{c,c'})$$

Hence, the matrix $\widetilde{U} \in \mathbb{R}^{\mathcal{C}^2 \times \mathcal{U}}$ with entries defined by

$$\widetilde{u}_{v,a} = \widetilde{\pi}(v) \cdot (\widetilde{\lambda}_v(a, 1) - \widetilde{\mu}_v(a, 1)) = -\widetilde{\pi}(v) \cdot (\widetilde{\lambda}_v(a, -1) - \widetilde{\mu}_v(a, -1))$$

satisfies

$$\widetilde{u}_{(c,c'),a} = \begin{cases} \tau u_{(c,c'),a}, & \text{if } (c, c') \in S \\ 0, & \text{otherwise.} \end{cases}$$

It is easy to see from the definition of $\gamma_2^*$ that this implies $\gamma_2^*(\widetilde{U}) \leq \tau\gamma_2^*(U) = \frac{\alpha\gamma_2^*(U)}{D \bullet U}$. $\square$

*Proof of Lemma 12.* The main observation is that, since $\lambda$ and $\mu$ share the same marginal on $\mathcal{U}$ but the labels are given by the functions $s$ and $-s$, for any hypothesis $h : \mathcal{C} \to \{\pm 1\}$ we have $L_\lambda(h) + L_\mu(h) = 1$. Therefore,

$$(L_\lambda(h) - L_\lambda(c')) + (L_\mu(h) - L_\mu(c)) = ((L_\lambda(h) + L_\mu(h)) - (1 - L_\mu(c')) - L_\mu(c)$$
$$= L_\mu(c') - L_\mu(c) > \alpha.$$

This implies that if $L_\lambda(h) - L_\lambda(c') \leq \frac{\alpha}{4}$, then $L_\mu(h) - L_\mu(c) > \frac{3\alpha}{4}$, as required.

Suppose now that $\mathcal{M}$ $(\alpha/4, \beta)$-learns $\mathcal{C}$ agnostically with $n$ samples. Let $A \subseteq \{\pm 1\}^{\mathcal{U}}$ be the set of hypotheses with loss at most $L_\lambda(c') + \alpha/4$ on $\lambda^n$. As we just showed, every hypothesis in $A$ has loss larger than $L_\mu(c) + 3\alpha/4$ under $\mu$. Since

$$\min_{c'' \in \mathcal{C}} L_\lambda(c'') \leq L_\lambda(c'), \qquad \min_{c'' \in \mathcal{C}} L_\mu(c'') \leq L_\mu(c),$$

it follows from the definition of agnostic learning that $\mathbb{P}[\mathcal{M}(\lambda^n) \in A] \geq 1 - \beta$, and $\mathbb{P}[\mathcal{M}(\lambda^n) \in A] \leq \beta$. Then, by the definition of total variation,

$$d_{\text{TV}}(\mathcal{M}(\lambda^n), \mathcal{M}(\mu^n)) \geq \mathbb{P}[\mathcal{M}(\lambda^n) \in A] - \mathbb{P}[\mathcal{M}(\mu^n) \in A] \geq 1 - 2\beta,$$

completing the proof of the lemma. $\square$

# D  Omitted Proofs for the Realizable Refutation Lower Bound

## D.1  Derivation of dual formulation

*Proof of Lemma 15.* Let $L_\mathcal{C} = \{G \in \mathbb{R}^{\mathcal{C} \times (\mathcal{U} \times \{\pm 1\})} : \gamma_2(G) \leq t\}$. Let $K_\mathcal{C}$ and $K_\mathcal{C}'$ be as defined by equations (9) and (10). By definition, $\eta(\mathcal{C}, \alpha) > t$ if and only if $L_\mathcal{C}$ and $K_\mathcal{C}'$ are disjoint.

Given some $U \in \mathbb{R}^{\mathcal{C} \times (\mathcal{U} \times \{\pm 1\})}$, we are interested in the quantities $\max\{U \cdot G : G \in L_\mathcal{C}\}$ and $\min\{U \cdot G : G \in K_\mathcal{C}'\}$. In particular, by the hyperplane separation theorem, since $L_\mathcal{C}$ and $K_\mathcal{C}'$ are convex and $L_\mathcal{C}$ is also compact, they are disjoint exactly when there exists some $U \in \mathbb{R}^{\mathcal{C} \times (\mathcal{U} \times \{\pm 1\})}$ such that

$$\max\{U \cdot G : G \in L_\mathcal{C}\} < \min\{U \cdot G : G \in K_\mathcal{C}'\}.$$

By definition,

$$\max\{U \cdot G : G \in L_\mathcal{C}\} = t\gamma_2^*(U).$$

Also,

$$\min\{U \cdot G : G \in K_\mathcal{C}'\}$$
$$= \min_{G \in K_\mathcal{C}'} \sum_{c \in \mathcal{C}, a \in \mathcal{U}} (u_{c,(a,c(a))} g_{c,(a,c(a))} + u_{c,(a,-c(a))} g_{c,(a,-c(a))})$$
$$= \min_{\substack{G \in K_\mathcal{C} \\ \theta \in \mathbb{R}^\mathcal{C}}} \sum_{c \in \mathcal{C}, a \in \mathcal{U}} (u_{c,(a,c(a))} \cdot (g_{c,(a,c(a))} + \theta_c) + u_{c,(a,-c(a))} \cdot (g_{c,(a,-c(a))} + \theta_c))$$
$$= \min_{G \in K_\mathcal{C}} \sum_{c \in \mathcal{C}, a \in \mathcal{U}} (u_{c,(a,c(a))} \cdot g_{c,(a,c(a))} + u_{c,(a,-c(a))} \cdot g_{c,(a,-c(a))})$$
$$+ \min_{\theta \in \mathbb{R}^\mathcal{C}} \sum_{c \in \mathcal{C}} \theta_c \cdot \sum_{a \in \mathcal{U}} (u_{c,(a,c(a))} + u_{c,(a,-c(a))})$$

If, for some $c \in \mathcal{C}$, it holds that $\sum_{a \in \mathcal{U}}(u_{c,(a,c(a))} + u_{c,(a,-c(a))}) \neq 0$, then

$$\min_{\theta_c \in \mathbb{R}} \theta_c \cdot \sum_{a \in \mathcal{U}} (u_{c,(a,c(a))} + u_{c,(a,-c(a))}) = -\infty.$$

Also, if there exist $c \in \mathcal{C}$ and $x \in \mathcal{U}$ such that $u_{c,(a,-c(a))} < 0$, then

$$\min_{G \in K_\mathcal{C}} u_{c,(a,-c(a))} g_{c,(a,-c(a))} = -\infty.$$

However, in the remaining case where $U$ is in the set $S_\mathcal{C}$, then

$$\min\{U \cdot G : G \in K_\mathcal{C}'\} = \min_{G \in K_\mathcal{C}} \sum_{c \in \mathcal{C}, a \in \mathcal{U}} (u_{c,(a,c(a))} \cdot g_{c,(a,c(a))} + u_{c,(a,-c(a))} \cdot g_{c,(a,-c(a))})$$
$$= \sum_{c \in \mathcal{C}, a \in \mathcal{U}} (-\alpha |u_{c,(a,c(a))}| + u_{c,(a,-c(a))}).$$

With these facts at our disposal, we obtain

$$\eta(\mathcal{C}, \alpha) > t \Leftrightarrow K_\mathcal{C}' \cap L_\mathcal{C} = \emptyset$$
$$\Leftrightarrow \exists U \in \mathbb{R}^{\mathcal{C} \times (\mathcal{U} \times \{\pm 1\})}, \ \max\{U \cdot G : G \in L_\mathcal{C}\} < \min\{U \cdot G : G \in K_\mathcal{C}\}$$
$$\Leftrightarrow \exists U \in S_\mathcal{C}, \ t\gamma_2^*(U) < \sum_{c \in \mathcal{C}, a \in \mathcal{U}} (-\alpha |u_{c,(a,c(a))}| + u_{c,(a,-c(a))})$$
$$\Leftrightarrow \max_{U \in S_\mathcal{C}} \frac{\sum_{c \in \mathcal{C}, a \in \mathcal{U}} (u_{c,(a,-c(a))} - \alpha |u_{c,(a,c(a))}|)}{\gamma_2^*(U)} > t$$

Since the equivalence holds for all $t \in \mathbb{R}$, it follows that

$$\eta(\mathcal{C}, \alpha) = \max_{U \in S_\mathcal{C}} \frac{\sum_{c \in \mathcal{C}, a \in \mathcal{U}} (u_{c,(a,-c(a))} - \alpha |u_{c,(a,c(a))}|)}{\gamma_2^*(U)}.$$

$\square$

## D.2 General case of realizable refutation lower bound

Section 3.1.3 proved our lower bound for realizable refutation in the single-concept case. In the general case, there are two issues to resolve:

1. Instead of equation (12) holding for each concept, it holds on average. In particular,

$$\sum_{c \in \mathcal{C}, a \in \mathcal{U}} u_{c,(a,-c(a))} > \sum_{c \in \mathcal{C}, a \in \mathcal{U}} \alpha |u_{c,(a,c(a))}|.$$

   Equivalently,

$$\mathbb{E}_{c \sim \pi}\left[L_\lambda(c)\right] > \frac{2\alpha}{1+\alpha}. \tag{14}$$

2. Even if we can guarantee for a concept $c \in \mathcal{C}$ that

$$L_{\lambda_c}(c) > \frac{2\alpha}{1+\alpha},$$

   it may hold, for some other $h : \mathcal{U} \to \{\pm 1\}$, that $L_{\lambda_c}(h)$ is small. We need to rule out this possibility in order to give a lower bound against refutation.

The first issue is resolved in Lemma 20 by applying the binning result of Lemma 19. The second issue will be resolved in Lemma 21.

**Lemma 20.** *Suppose there exist families $\{\lambda_c\}_{c \in \mathcal{C}}$ and $\{\mu_c\}_{c \in \mathcal{C}}$ of distributions over $\mathcal{U}$, together with a parameter distribution $\pi$ over $\mathcal{C}$, such that*

$$\Delta = \mathbb{E}_{c \sim \pi}\left[L_{\lambda_c}(c)\right] > \frac{2\alpha}{1+\alpha}$$

*while, for all $c \in \mathcal{C}$, $L_{\mu_c}(c) = 0$. Let, further, $U \in \mathbb{R}^{\mathcal{C} \times \mathcal{U}}$ be the matrix with entries $u_{c,a} = \pi(c)(\lambda_c(a) - \mu_c(a))$.*

*Then there exist families $\{\widetilde{\lambda}_c\}_{c \in \mathcal{C}}$ and $\{\widetilde{\mu}_c\}_{c \in \mathcal{C}}$ of distributions over $\mathcal{U} \times \{\pm 1\}$, together with a parameter distribution $\widetilde{\pi}$ over $\mathcal{C}$, such that, for all $c$ in the support of $\widetilde{\pi}$,*

$$L_{\widetilde{\lambda}_c}(c) \geq \Omega\left(\frac{\alpha}{1+\alpha} \bigg/ \log\left(\frac{1+\alpha}{\alpha}\right)\right),$$

*while still $L_{\widetilde{\mu}_c}(c) = 0$ for all $c \in \mathcal{C}$. Moreover, the matrix $\widetilde{U} \in \mathbb{R}^{\mathcal{C} \times \mathcal{U}}$ with entries $\widetilde{u}_{c,a} = \widetilde{\pi}(c)(\widetilde{\lambda}_c(a) - \widetilde{\mu}_c(a))$ satisfies*

$$\gamma_2^*(\widetilde{U}) \leq \frac{2\alpha \gamma_2^*(U)}{(1+\alpha)\Delta}.$$

*Proof.* Apply Lemma 19, with $a_c = L_{\lambda_c}(c)$ for all $c \in \mathcal{C}$, and $\beta = \frac{\alpha}{1+\alpha} < \frac{\Delta}{2}$, to obtain $S \subseteq \mathcal{C}$ such that

$$\pi(S) \cdot \min_{c \in S} a_c \geq \frac{\Delta - \beta}{O(\log(1/\beta))} \geq \frac{\Delta}{O(\log((1+\alpha)/\alpha))}.$$

Let $\widetilde{\pi}$ be $\pi$ conditional on membership in $S$. Thus,

$$\widetilde{\pi}(v) = \begin{cases} \pi(v)/\pi(S), & \text{if } v \in S \\ 0, & \text{otherwise.} \end{cases}$$

Let $\tau = \frac{2\alpha}{(1+\alpha)\Delta} \in (0,1)$. For all $c \in \mathcal{C}$, define $\widetilde{\mu}_c = \mu_c$ and $\widetilde{\lambda}_c = \tau\pi(S)\lambda_c + (1 - \tau\pi(S))\mu_c$. Then, for all $c$ in the support of $\widetilde{\pi}$,

$$L_{\widetilde{\mu}_c}(c) = L_{\mu_c}(c) = 0$$

$$L_{\widetilde{\lambda}_c}(c) = \tau \cdot \pi(S) \cdot L_{\lambda_c}(c) \geq \frac{\frac{\alpha}{1+\alpha}}{O\left(\left(\log\left(\frac{1+\alpha}{\alpha}\right)\right)\right)}.$$

Moreover, the matrix $\widetilde{U} \in \mathbb{R}^{\mathcal{C} \times \mathcal{U}}$ with entries $\widetilde{u}_{c,a} = \widetilde{\pi}(v)(\widetilde{\lambda}_v(a) - \widetilde{\mu}_v(a))$ is obtained from the matrix $\tau U$ by replacing some of its rows with the zero-vector. It follows immediately that $\gamma_2^*(\widetilde{U}) \leq \tau\gamma_2^*(U) = \frac{2\alpha\gamma_2^*(U)}{(1+\alpha)\Delta}$. $\qquad\square$

**Lemma 21.** *Suppose we have distributions $\lambda_c$ and $\mu_c$ on $\mathcal{U} \times \{\pm 1\}$ for each $c \in \mathcal{C}$ where:*

*(a)* $L_{\mu_c}(c) = 0$;

*(b)* $L_{\lambda_c}(c) > \alpha$.

*Then there exist distributions $\widetilde{\lambda}_c$ and $\widetilde{\mu}_c$ for each $c \in \mathcal{C}$ such that:*

*(c)* $L_{\widetilde{\mu}_c}(c) = 0$;

*(d)* $\forall h : \mathcal{U} \to \{\pm 1\}, \ L_{\widetilde{\lambda}_c}(h) > \frac{\alpha}{2}$;

*(e)* $\widetilde{\lambda}_c - \widetilde{\mu}_c = \frac{1}{2}(\lambda_c - \mu_c)$.

*Proof.* For $c \in \mathcal{C}$, let $\sigma_c$ be the distribution on $\mathcal{U} \times \{\pm 1\}$ which has the same marginal on $\mathcal{U}$ as does $\lambda_c$, and which satisfies $c(a) = b$ for all $(a, b)$ in the support of $\sigma_c$. Also, let $\widetilde{\lambda}_c = \frac{1}{2}\lambda_c + \frac{1}{2}\sigma_c$ and $\widetilde{\mu}_c = \frac{1}{2}\mu_c + \frac{1}{2}\sigma_c$. Properties (c) and (e) follow immediately.

To establish property (d), notice first that, for any $a \in \mathcal{U}$ in the support of $\lambda_c$, $\mathbb{P}_{\widetilde{\lambda}_c}[b = c(a) \mid a] \geq \frac{1}{2}$, and also $\mathbb{P}_{\widetilde{\lambda}_c}[b \neq c(a) \mid a] = \frac{1}{2} \cdot \mathbb{P}_{\lambda_c}[b \neq c(a) \mid a]$. Then, for any function $h : \mathcal{U} \to \{\pm 1\}$.

$$
\begin{aligned}
L_{\widetilde{\lambda}_c}(h) &= \mathbb{P}_{\widetilde{\lambda}_c}[h(a) \neq b] \\
&= \sum_{(a,b) \in \mathcal{U} \times \{\pm 1\}} \mathbb{P}_{\widetilde{\lambda}_c}[a] \cdot \mathbb{P}_{\widetilde{\lambda}_c}[h(a) \neq b \mid a] \\
&\geq \sum_{(a,b) \in \mathcal{U} \times \{\pm 1\}} \mathbb{P}_{\widetilde{\lambda}_c}[a] \cdot \min\left\{ \mathbb{P}_{\widetilde{\lambda}_c}[b = c(a) \mid a], \mathbb{P}_{\widetilde{\lambda}_c}[b \neq c(a) \mid a] \right\} \\
&\geq \sum_{(a,b) \in \mathcal{U} \times \{\pm 1\}} \mathbb{P}_{\lambda_c}[a] \cdot \min\left\{ \frac{1}{2}, \frac{1}{2} \cdot \mathbb{P}_{\lambda_c}[b \neq c(a) \mid a] \right\} \\
&= \frac{1}{2} \cdot \sum_{(a,b) \in \mathcal{U} \times \{\pm 1\}} \mathbb{P}_{\lambda_c}[a] \cdot \mathbb{P}_{\lambda_c}[b \neq c(a) \mid a] \\
&= \frac{1}{2} \cdot L_{\lambda_c}(c) \\
&> \frac{\alpha}{2}. \qquad \qquad \qquad \square
\end{aligned}
$$

Equipped with Lemmas 20 and 21, we are ready to prove our lower bound against realizable refutation.

*Proof of Theorem 2.* We define the parameter distribution $\pi$ over $\mathcal{C}$, and the distribution families $\{\lambda_c\}_{c \in \mathcal{C}}$ and $\{\mu_c\}_{c \in \mathcal{C}}$ over $\mathcal{U} \times \{\pm 1\}$, as in Section 3.1.2. We denote $\Delta = \mathbb{E}_{c \sim \pi}[L_{\lambda_c}(c)]$. By equation (14), together with Lemmas 20 and 21, we obtain modified families of distributions $\{\widetilde{\lambda}_c\}_{c \in \mathcal{C}}$ and $\{\widetilde{\mu}_c\}_{c \in \mathcal{C}}$, together with a parameter distribution $\widetilde{\pi}$ over $\mathcal{C}$, such that, for all $c$ in the support of $\widetilde{\pi}$, and for all functions $h : \mathcal{U} \to \{\pm 1\}$,

$$
L_{\widetilde{\lambda}_c}(h) = \Omega\left( \frac{\alpha}{1 + \alpha} \middle/ \log\left( \frac{1 + \alpha}{\alpha} \right) \right)
$$

while $L_{\widetilde{\mu}_c}(c) = 0$ for all $c \in \mathcal{C}$. By Lemmas 11, 20 and 21, we may assume further that the matrix $\widetilde{M} \in \mathbb{R}^{\mathcal{C} \times (\mathcal{U} \times \{\pm 1\})}$ with entries $m_{c,(a,b)} = \widetilde{\lambda}_c(a, b) - \widetilde{\mu}_c(a, b)$ satisfies

$$
\|\widetilde{M}\|_{\ell_\infty \to L_2(\widetilde{\pi})} \leq \frac{4\alpha \gamma_2^*(U)}{(1 + \alpha)\Delta}.
$$

Now let $\mathcal{M}$ be an $\varepsilon$-LDP protocol which is able to distinguish a labeling by some $c \in \mathcal{C}$ from a distribution with which every function $h : \mathcal{U} \to \{\pm 1\}$ disagrees with the labels with probability $\Omega\left( \frac{\alpha}{1 + \alpha} \middle/ \log\left( \frac{1 + \alpha}{\alpha} \right) \right)$. If this is true, then, for every $c \in \mathcal{C}$ in the support of $\widetilde{\pi}$,

$$
D_{\mathrm{KL}}(\mathcal{T}_{\mathcal{M}}(\widetilde{\lambda}_c^n) \| \mathcal{T}_{\mathcal{M}}(\widetilde{\mu}_c^n)) = \Omega(1).
$$

Meanwhile, Lemma 8 guarantees

$$\mathop{\mathbb{E}}_{c\sim\widetilde{\pi}}\left[\mathrm{D}_{\mathrm{KL}}(\mathcal{T}_{\mathcal{M}}(\widetilde{\lambda}_c^n)\|\mathcal{T}_{\mathcal{M}}(\widetilde{\mu}_c^n))\right] \le O(n\varepsilon^2)\cdot\|\widetilde{M}\|_{\ell_\infty\to L_2(\pi)}^2,$$

whereby we obtain

$$n = \Omega\left(\frac{1}{\varepsilon^2\cdot\|\widetilde{M}\|_{\ell_\infty\to L_2(\pi)}^2}\right) = \Omega\left(\frac{(1+\alpha)^2\Delta^2}{\varepsilon^2\alpha^2\gamma_2^*(U)^2}\right). \tag{15}$$

Now we may use

$$\gamma_2^*(U) = \frac{\sum_{c\in\mathcal{C},a\in\mathcal{U}}(u_{c,(a,-c(a))} - \alpha|u_{c,(a,c(a))}|)}{\eta(\mathcal{C},\alpha)}. \tag{16}$$

Note that, for any $c\in\mathcal{C}$, since $L_{\mu_c}(c) = 0$,

$$\frac{1}{\pi(c)}\sum_{a\in\mathcal{U}} u_{c,(a,-c(a))} - \alpha|u_{c,(a,c(a))}|$$
$$= \mathbb{P}_{(a,b)\sim\lambda_c}[c(a)\ne b] - \mathbb{P}_{(a,b)\sim\mu_c}[c(a)\ne b]$$
$$\quad - \alpha\cdot\left(\mathbb{P}_{(a,b)\sim\lambda_c}[c(a)=b] + \mathbb{P}_{(a,b)\sim\mu_c}[c(a)=b]\right)$$
$$= (1+\alpha)\cdot\mathbb{P}_{(a,b)\sim\lambda_c}[c(a)\ne b] - 2\alpha$$
$$= (1+\alpha)\cdot L_{\lambda_c}(c) - 2\alpha.$$

Taking expectations over $c\sim\pi$, we have

$$\sum_{c\in\mathcal{C},a\in\mathcal{U}} u_{c,(a,-c(a))} - \alpha|u_{c,(a,c(a))}| = (1+\alpha)\cdot\mathop{\mathbb{E}}_{c\sim\pi}[L_{\lambda_c}(c)] - 2\alpha$$
$$= (1+\alpha)\cdot\Delta - 2\alpha. \tag{17}$$

Putting equations (15), (16), and (17) together, we have

$$n = \Omega\left(\frac{(1+\alpha)^2\Delta^2\eta(\mathcal{C},\alpha)^2}{\varepsilon^2\alpha^2((1+\alpha)\Delta-2\alpha)^2}\right) = \Omega\left(\frac{\eta(\mathcal{C},\alpha)^2}{\varepsilon^2\alpha^2}\right). \qquad\qquad \square$$

# E  Open problems

This work, together with Edmonds et al. [2019], largely completes the picture of agnostic refutability and learnability under non-interactive LDP. In the realizable setting, we have shown that refutation implies learning for non-interactive LDP. It is an interesting open problem to determine the converse – whether realizable learning implies refutation. Secondly, for an arbitrary concept class $\mathcal{C}$, can we obtain a characterization of realizable learnability in terms of a quantity which is efficiently computable from the definition of $\mathcal{C}$? Furthermore, for both the realizable and agnostic versions, the relationships obtained between the sample complexities of refutability and learnability in this work are indirect, via characterizations of these tasks by the approximate $\gamma_2$ norm. For example, although realizable refutability implies realizable learnability under non-interactive LDP, it remains open how one might obtain a non-interactive LDP protocol for learnability directly from one for refutability.