# OpenReview forum: "On Learning and Refutation in Noninteractive Local Differential Privacy"
_NeurIPS.cc/2022/Conference — NeurIPS 2022 Accept_

### Official Review · Reviewer_eniG · 2022-07-10

**Rating:** 6
**Confidence:** 2
**Soundness:** 3 good
**Presentation:** 3 good
**Contribution:** 3 good

**Summary:**

This paper studies the sample complexity of learning and refutation in the model of non-interactive local differential privacy (LDP). In a non-interactive LDP protocol, all users sends a message to the server once, simultaneously and independently. In the agnostic setting, the authors show that learning and refutation are equivalent (up to a logarithmic approximation) wrt sample complexity. The main result gives a lower bound on the sample complexity in terms of the "approximate $\gamma_2$ norm" of a matrix associated with the concept class, which in conjunction with earlier work by Edmonds et al. (2019) shows this equivalence. In the realizable setting, the authors show that refutation implies learning.


**Questions:**

* Why is the approximate $\gamma_2$ norm not defined in the main text? Given that this norm features extensively, a definition in the main text is desirable.

* In Lemma 9,  the $\bullet$ operator is not defined.

* Why is the realizable refutation setting more difficult than the agnostic case?

* What is a good intuition for the definition of the quantity $\eta(\mathcal{C},\alpha)$ in Definition 13 that features in the bounds in Theorem 2 for the realizable case?





**Limitations:**

The authors do not address any potential negative societial impact of the work and I reckon there is none.

**Strengths And Weaknesses:**

The paper resolves an open question for lower bounds in the agnostic case in the non-interactive LDP setting. The presentation in certain parts can be improved. In particular, deferring some of the full proofs to the appendix and including some intuition for the two main quantities that appears in the statements of theorems, viz., $\gamma_2$ and $\eta$ can make the presentation better. Also the reader can benefit from a separate Contributions section with pointers to the main statements.

---

> ### Author Response · Authors · 2022-08-02
> **Response to reviewer eniG**
>
> We thank the reviewer for their interest in our paper.
>
> We have added a definition of the approximate $\gamma_2$ norm to the main text, and a definition of the matrix (i.e., Hadamard) dot product operator
>
> > Why is the realizable refutation setting more difficult than the agnostic case?
>
> Realizable refutation and learning are special cases of agnostic refutation which impose additional structure on the problem i.e., realizability of the data distribution. This poses a practical hurdle in obtaining lower bounds against realizable learning and refutation, as the construction of hard distributions in a lower bound proof needs to respect realizability. This issue is most pronounced in the case of realizable learning, where there is no requirement on how the learning algorithm must behave on non-realizable distributions, so the lower bound argument must use only realizable distributions. At present, none of our methods give lower bounds against realizable learning. The pairs of distributions in our lower bound against agnostic learning are not guaranteed to be realizable. Moreover, in some cases the sample complexity of agnostic learning in the non-interactive LDP model is provably much larger than the sample complexity of realizable learning, e.g., for learning k-wise disjunctions. This means that the approximate $\gamma_2$ norm cannot characterize realizable learning. Similarly, our proof of the realizable refutation lower bound constructs pairs of hard distributions, one of which agrees with a concept while the other is not well-approximated by any hypothesis. Since the second distribution is not realizable, there is no guarantee how a realizable learner behaves on it. At present, we do not know if $\eta(\mathcal{C},\alpha)$ characterizes realizable learning in the non-interactive LDP model.
>
> The only method we know of that gives lower bounds for realizable learning tailored to non-interactive LDP is the margin complexity lower bound of Daniely and Feldman. Unfortunately, margin complexity can sometimes be exponentially smaller than the sample complexity of realizable learning in this model, as shown by Dagan and Feldman.
>
> > What is a good intuition for the definition of the quantity $\eta(\mathcal{C},\alpha)$  in Definition 13 that features in the bounds in Theorem 2 for the realizable case?
>
> Consider the matrix $W$ in the set $K_\mathcal{C}$, as given by equation (9). Answering the statistical queries associated with $W$ may be thought of as estimating a surrogate loss function for each concept in the concept class. This surrogate loss is guaranteed to be close to zero on points which agree with the concept, and at least 1 on points which disagree with the concept. In particular, the total surrogate loss will be close to zero on realizable distributions, while it is large on distributions which disagree with the concept with large probability.
> Meanwhile, each matrix the set $K_{\mathcal{C}}'$, defined by equation (10), is obtained by shifting each row of some matrix $W$ from $K_\mathcal{C}$ by a row-specific constant. This has the effect of adding a constant to the surrogate losses given by $W$, which can be easily recovered by subtraction. Allowing this shift can potentially decrease the $\gamma_2$ norm, and, therefore, the sample complexity of the algorithm.
>
> We have tried to convey this intuition in the revised submission.

---

### Official Review · Reviewer_d2S2 · 2022-07-11

**Rating:** 8
**Confidence:** 3
**Soundness:** 4 excellent
**Presentation:** 4 excellent
**Contribution:** 3 good

**Summary:**

Setting and main contributions:
The paper is about the tasks of learning and refutation concept classes in the local differential privacy (LDP) model (which is a model with strong privacy guarantees for users, even from the central party).

The first main result is closing an open question of showing a lower bound on the sample complexity for agnostic learning. This gives a full characterization of these models, as upper bounds for learning and refutation, and a lower bound for refutation were obtained by Edmonds, Nikolov, and Ullman (STOC20).

The second main contribution is in the realizable model: an LDP protocol with upper bounds on the sample complexity for learning and refutation, and also a lower bound on refutation.

**Questions:**

-

**Ethics Review Area:**

["I don’t know"]

**Limitations:**

-

**Strengths And Weaknesses:**

The paper studies fundamental questions regarding learning for the important privacy model - LDP.
The contributions are significant.

The first contribution closes the gap in the literature on learning and refutation in the agnostic model.
Specifically, I find the technique for proving the lower bound very elegant.
The idea of working with the difference matrix seems interesting and may be of independent interest, in my opinion.

In the second contribution, the authors define a quantity, eta(C,alpha), that quantifies the sample complexity in the realizable case. The lower bound idea is similar to the agnostic case when working with the "right" quantity eta(C,alpha).

I recommend accepting the paper.

---

> ### Author Response · Authors · 2022-08-02
> **Response to reviewer d2S2**
>
> We thank the reviewer for their interest in our paper.

---

### Official Review · Reviewer_kF1C · 2022-07-12

**Rating:** 7
**Confidence:** 2
**Soundness:** 3 good
**Presentation:** 3 good
**Contribution:** 3 good

**Summary:**

This paper studies two tasks, learning and refutation, in non-interactive local differential privacy. It proves the same lower bounds of the sample complexity of agnostic PAC learning for both tasks, which matches the upper bound proved in the previous work. Consequently, the paper shows the equivalence between learning and refutation in the agnostic setting.

**Questions:**

As the paper is studied for the finite concept class, one question is whether the result could be extended to the infinite concept class setting.

**Limitations:**

Yes, the paper describes several open problems as the future work.

**Strengths And Weaknesses:**

### Strengths
The result is solid and complete. It closes the gap between upper bound and lower bound of sample complexity for learning and refutation, in the setting of agnostic PAC learning and non-interactive local differential privacy.

### Weaknesses
The proof could be more hierarchical.

Minor issue:
1. Typo: In lemma 8, m_{v, x} -> m_{a, x}

---

> ### Author Response · Authors · 2022-08-02
> **Response to reviewer kF1C**
>
> We thank the reviewer for their interest in our paper.
>
> At this point we do not know how to extend our upper bounds to infinite concept classes, and this remains an important open question for future work. Our lower bounds can be applied to any finite subset of the concept class.

---

### Official Review · Reviewer_3PmZ · 2022-07-13

**Rating:** 7
**Confidence:** 2
**Soundness:** 3 good
**Presentation:** 3 good
**Contribution:** 3 good

**Summary:**

This paper studies the issue of learning and refutation in the setting of non-interactive local differential privacy. It provides a characterization of the sample complexity of agnostic PAC learning for non-interactive LDP protocols and shows the equivalence of learning and refutation in the setting of agnostic learning.


**Questions:**

 I have some one major concern:

(1)What is the relation of the refutation in this paper to that in Vadhan [2017]? I found that they are different but could not see the direct equivalence.


**Limitations:**

It needs some more detailed explanation of the relationship to the similar issues in the literature.

**Strengths And Weaknesses:**

I am not an expert in this field.  I believe that the results in this paper are mathematically sound.  The presentation is well-organized and also the topic about learning and refutation in the LDP setting is very interesting.

One minor: some mistakes in the definition of learning problem Page 1

---

> ### Author Response · Authors · 2022-08-02
> **Response to reviewer 3PmZ**
>
> We thank the reviewer for their interest in our paper. Let us briefly explain the relationship between our definition of refutation and definitions in prior work.
>
> Vadhan’s definition of random right-hand side refutation requires that the refutation algorithm satisfies the following properties:
> * For any sequence of $n$ labeled samples $(x_1, b_1), \ldots, (x_n, b_n)$ that is correctly labeled by some concept in $\mathcal{C}$ (i.e., for some $c \in \mathcal{C}$, $c(x_i) = b_i$ for all $i$), then the algorithm outputs “random” with probability less than 1/3.
> * For any sequence of $n$ labeled samples $x_1,, \ldots, x_n$ and labels $b_1, \ldots b_n$ that are sampled uniformly and independently from $\{-1, +1\}$, the algorithm outputs “random” with probability at least 2/3.
>
> A minor difference with our definition of realizable refutation is that Vadhan’s definition is stated with respect to any sequence of $n$ samples, whereas our definitions are with respect to any distribution $\lambda$ on samples. The more important difference is that our definition has an additional parameter $\alpha$, and requires the refutation algorithm to output “random” when the loss of any concept (i.e. any function $h:\mathcal{U} \to \\{-1, +1\\}$) is at least $\alpha$. In the special case when $\alpha = 1/2$ this coincides with Vadhan’s definition, since the loss of every concept is $1/2$ only when each sample is labeled with a uniformly random label. In Vadhan’s paper, refutation at $\alpha = 1/2$ is sufficient to imply learning to within any accuracy, because of boosting. We are not aware of a boosting algorithm that works in non-interactive local differential privacy, i.e., that does not require additional interaction.
>
> The only difference between our definition of agnostic refutation and the definition of refutation in Kothari and Livni’s work is that our definition is distribution-free, whereas they define the refutation and learning problems for fixed distributions.
>
> In terms of results, both Vadhan, and Kothari and Livni show that refutation (in their respective formulations) is equivalent to learning (respectively, realizable and agnostic). Their results do not apply in the non-interactive local model of differential privacy because their reductions,  both from learning to refutation, and from refutation to learning, require more than one round of interaction. We recover and analog of Kothari and Livni’s result in the non-interactive setting, as well as one direction (refutation implies learning) of Vadhan’s result.
>
> We have added a brief explanation of these distinctions in the appendix of our revised submission.

---

### Author Response · Authors · 2022-08-02
**Response to initial reviews**

We thank all reviewers for their time and their thoughtful comments. We have submitted a revision of the paper, in which we fixed several typos, and did our best to address reviewer comments within the space limits.

---

### Meta-Review · Area_Chair_edJk · 2022-08-26

**Recommendation:** Accept
**Confidence:** Certain

**Metareview:**

This paper provides a characterization of the sample complexity of agnostic learning under local differential privacy (LDP) and show that it is equivalent to refutation. The mathematical insights from the paper might be valuable in future research. It is a clean contribution, with positive feedback from all reviewers.

**Award:**

No

---

### Decision · Program_Chairs · 2022-09-14

Accept